# A safety mechanism enables tissue-specific resistance to protein aggregation during aging in *C. elegans*

**Raimund Jung[1], Marie C. Lechler[1,2], Ana Fernandez-Villegas[3], Chyi Wei Chung[3¤], Harry C. Jones[4], Yoon Hee Choi[4], Maximilian A. Thompson[4], Christian Rödelsperger[5], Waltraud Röseler[5], Gabriele S. Kaminski Schierle[3], Ralf J. Sommer[5], Della C. David** [1,4,6]*

**1** German Center for Neurodegenerative Diseases (DZNE), Tübingen, Germany, **2** Graduate Training Centre of Neuroscience, International Max Planck Research School, Tübingen, Germany, **3** Chemical Engineering and Biotechnology, University of Cambridge, Cambridge, United Kingdom, **4** The Babraham Institute, Signalling Program, Cambridge, United Kingdom, **5** Max Planck Institute for Developmental Biology, Department for Integrative Evolutionary Biology, Tübingen, Germany, **6** Interfaculty Institute of Biochemistry, University of Tübingen, Tübingen, Germany

¤ Current address: Kavli Nanoscience Institute for Nanoscience Discovery, Department of Physics, University of Oxford, Oxford, United Kingdom

* della.david@babraham.ac.uk

**Data Availability Statement:** All relevant data are within the paper and its Supporting Information files. All sequencing data has been submitted to the

## Abstract

During aging, proteostasis capacity declines and distinct proteins become unstable and can accumulate as protein aggregates inside and outside of cells. Both in disease and during aging, proteins selectively aggregate in certain tissues and not others. Yet, tissue-specific regulation of cytoplasmic protein aggregation remains poorly understood. Surprisingly, we found that the inhibition of 3 core protein quality control systems, namely chaperones, the proteasome, and macroautophagy, leads to lower levels of age-dependent protein aggregation in *Caenorhabditis elegans* pharyngeal muscles, but higher levels in body-wall muscles. We describe a novel safety mechanism that selectively targets newly synthesized proteins to suppress their aggregation and associated proteotoxicity. The safety mechanism relies on macroautophagy-independent lysosomal degradation and involves several previously uncharacterized components of the intracellular pathogen response (IPR). We propose that this protective mechanism engages an anti-aggregation machinery targeting aggregating proteins for lysosomal degradation.

## Introduction

Active mechanisms to ensure that proteins are functional and undamaged are crucial across lifeforms [1]. A healthy proteome is maintained both inside and outside of cells by a complex network of proteostasis components [2,3]. However, with age, disruption of protein folding and proteolytic activities leads to a decline in proteostasis capacity [3]. Inhibition of protein quality control (PQC) systems and acute stress such as heat shock cause protein instability and ultimately protein aggregation [4–7]. Aberrant protein deposition with highly structured

European Nucleotide archive under the study accession PRJEB41493.

**Funding:** This work was supported by funding from the Deutsches Zentrum für Neurodegenerative Erkrankungen (DZNE, German Center for Neurodegenerative Diseases), a Marie Curie International Reintegration Grant (FP7 People, 322120), an Institute Strategic Programme Grant (BB/P013384/1) from the Biotechnology and Biological Sciences Research Council to DCD. DCD was supported in part by the Deutsche Forschungsgemeinschaft (DFG, German Research Foundation, DA 1906/4-1) and HCJ by a BBSRC Babraham Research Campus Studentship (R1615-B0870). Work by WR, CR and RJS was supported by the Max Planck Society through funds to RJS. CWC received PhD funding from the Cambridge Trust (Cambridge Commonwealth, European & International Trust) and Wolfson College. GSKS acknowledges funding from the Wellcome Trust (065807/Z/01/Z) (203249/Z/16/Z), the UK Medical Research Council (MRC) (MR/K02292X/1), Alzheimer Research UK (ARUK) (ARUK-PG013-14), Michael J Fox Foundation (16238 and 022159) and Infinitus China Ltd. The funders had no role in study design, data collection and analysis, decision to publish, or preparation of the manuscript.

**Competing interests:** The authors have declared that no competing interests exist.

**Abbreviations:** ac, after conversion; bc, before conversion; GOMED, Golgi membrane-associated degradation; HRP, horseradish peroxidase; HSF-1, heat shock factor 1; IPR, intracellular pathogen response; KIN-19, casein kinase I isoform alpha; OE, overexpression; PAB-1, polyadenylate-binding protein 1; PCR, polymerase chain reaction; PQC, protein quality control; RHO-1, Ras-like GTP-binding protein rhoA; RNAi, RNA interference; SAPA, safeguard against protein aggregation; TCSPC, time-correlated single photon counting.

amyloid fibrils is a prominent pathological feature in age-related diseases such as neurodegenerative disorders, amyloidoses, and type II diabetes [8]. Also during normal aging with the concomitant decline in proteostasis, the intrinsic aggregation propensity of certain proteins becomes a challenge for the organism [9–11] and widespread protein aggregation is found in aged animals of different species, in the absence of disease processes [5,12–20]. Age-dependent aggregates show similarities to disease-associated protein aggregates and contain amyloid-like structures [17,21]. Notably, evidence from the model organism *Caenorhabditis elegans* reveals that age-dependent aggregates are proteotoxic and accelerate the functional decline of tissues [21,22]. Moreover, age-dependent aggregates can initiate disease-related protein aggregation [15,23].

Certain tissues and cell types are more likely to accumulate aggregates than others. In the early stages of neurodegenerative disorders, selective vulnerability to protein aggregation in distinct brain regions is particularly striking despite similar levels of expression of the disease-associated protein in less-susceptible brain regions [24,25]. Emerging evidence highlights cell and tissue specificity also in the age-dependent aggregation process [14,21,26,27]. However, what drives selective vulnerability or resilience to protein aggregation remains poorly understood [24,25]. One possible explanation is that different cells and tissues employ divergent proteostasis strategies. In particular, evidence from *C. elegans* reveals that tissues rely to varying degrees on the chaperone system, proteasome or macroautophagy degradation to alleviate protein misfolding in an age-dependent manner [28–31]. In addition to conventional PQC, specialized mechanisms monitor the aggregation process, such as active sequestration of aggregating proteins into compartments [32–34], disassembly by specialized chaperones [35–37], selective degradation of aggregates by macroautophagy (aggrephagy) [38], or expulsion in exophers [39].

Most of our current knowledge about the regulation of protein aggregation has been gained by examining disease-associated aggregating proteins as well as a few ectopically expressed standard aggregation-prone proteins such as luciferase. Moreover, compared to aggregating proteins with disordered regions [22,40], little is known about proteostasis mechanisms specifically targeting inherently aggregation-prone globular proteins [21]. Expanding research into age-dependent protein aggregation and its regulation in different tissues offers the opportunity to identify novel mechanisms to counteract selective vulnerability to protein aggregation. Here, we report the discovery of a tissue-specific mechanism that actively prevents protein aggregation in *C. elegans* pharyngeal muscles but not in the body-wall muscles. This safety mechanism is triggered in response to defective conventional PQC and it is highly effective at preventing the aggregation of globular proteins, KIN-19 and RHO-1. We found that aggregation is curbed by recognizing and eliminating newly synthesized proteins before their assembly into large aggregates. To prevent aggregation specifically in the pharynx, the safety mechanism relies on macroautophagy-independent lysosomal degradation and uncharacterized factors previously connected to the host's response to natural intracellular pathogens affecting the digestive tract. Notably, by limiting protein aggregation, the safety mechanism alleviates proteotoxicity.

## Results

### Aggregation of KIN-19 and RHO-1 is reduced in pharyngeal muscles in response to PQC impairment

To identify potential tissue-specific mechanisms controlling age-dependent protein aggregation, we investigated the role of PQC in 2 different tissues in *C. elegans*: the pharyngeal muscles, which are non-striated, and the body-wall muscles, which are striated. Well before old

age, *C. elegans* experiences functional decline in both tissues, potentially at different rates [41,42]. To evaluate changes in protein aggregation in response to PQC impairment, we examined fluorescent-tagged Ras-like GTP-binding protein rhoA (RHO-1) in pharyngeal muscles and casein kinase I isoform alpha (KIN-19) aggregation in the pharynx and body-wall muscles. Both globular proteins are highly prone to aggregate with age, becoming insoluble in strong detergents and redistributing into compact immobile puncta as assessed by the absence of fluorescence recovery after photobleaching [17,21,22]. Aggregates formed by these proteins display amyloid-like structures similar to disease-associated protein aggregates [21]. In contrast to KIN-19, when RHO-1 is overexpressed in the pharynx, it aggregates abundantly already in young animals. We visually assessed a population of adult transgenic *C. elegans* over several days to quantify changes in aggregation in response to knocking down the 3 canonical core PQC systems: chaperone-mediated folding, degradation by the proteasome and macroautophagy. Specifically, we used well-characterized interventions by targeting the heat shock factor 1 (HSF-1), the main transcription factor controlling chaperone expression in *C. elegans* [43–45], proteasome subunits of the 20S core and 19S cap (PBS-3, PAS-6, RPT-6) [46–48] and 2 essential macroautophagy components involved in the formation of autophagosome precursors (ATG-18, the ortholog of human WIPI1 and WIPI2 as well as UNC-51, the ortholog of human ULK1 and ULK2) [30,49–51]. To our surprise, we observed opposite outcomes for protein aggregation in the different muscle types: We found that PQC inhibition accelerated KIN-19 aggregation with age in the body-wall muscles (Fig 1A and 1B). In contrast, PQC inhibition led to a significant reduction in RHO-1 and KIN-19 aggregation in the pharyngeal muscles in both young animals (2 to 4 days of adulthood) and during the course of aging in post-reproductive animals (6 to 10 days of adulthood) (Figs 1C–1F and S1A–S1D and S1 Data). Results with RNAi were replicated using loss of function mutations in *atg-18*, *unc-51*, and *hsf-1* (Figs 1C–1F and S1D) and by chemical inhibition of proteasome activity (S1E Fig). Of note, HSF-1 controls the expression of a wide array of chaperones, with and without heat stress [43,52]. In the absence of heat stress, Brunquell and colleagues detected a significant reduction of 11 heat shock protein (*hsp* annotated) genes, including *hsp-1*, in *C. elegans* exposed to *hsf-1* RNAi [43]. Similarly, RNA sequencing in the present study showed significantly reduced levels of *hsp-16.49*, *hsp-16.41*, and *hsp-16.2* as observed in Brunquell and colleagues, as well as reduced levels of *hsp-70* upon *hsf-1* RNAi compared to control RNAi, in at least 2 sample comparisons (S2 Data). In addition to classical chaperones, loss of *hsf-1* leads to the down-regulation of a diverse set of genes ([43], S2 Data). We investigated whether inhibition of a single major chaperone such as HSP-1 was sufficient to modulate protein aggregation in the pharynx and replicate the results obtained with *hsf-1* RNAi. Importantly, we observed reduced aggregation of both KIN-19 and RHO-1 upon treatment with *hsp-1* RNAi (S1F and S1G Fig). To verify that RNAi is effective in the pharynx, we showed that RNAi targeting *gfp* reduces pharyngeal GFP levels to below 25% of control levels already at day 1 (S2A Fig). Macroautophagy disruption by *atg-18* RNAi treatment was confirmed in the pharynx as previously shown by reduced autophagic flux resulting in an increase of autophagosome to autolysosome ratio (S2B Fig, [30]). Increased levels of ubiquitination reflect the effectiveness of the proteasome inhibition by RNAi (S2C Fig). Next, we tested how increasing core PQC capacity influences protein aggregation in the pharyngeal muscles. We found that both HSF-1 overexpression and enhancing proteasome activity by RPN-6 overexpression [53] alleviated pharyngeal protein aggregation (S3A and S3B Fig). Thus, the canonical core PQC systems are important for limiting protein aggregation in the pharynx. Together, these results raise the possibility that a mechanism is triggered in pharyngeal muscle cells to counteract local vulnerability to protein aggregation caused by severe and systemic PQC inhibition.

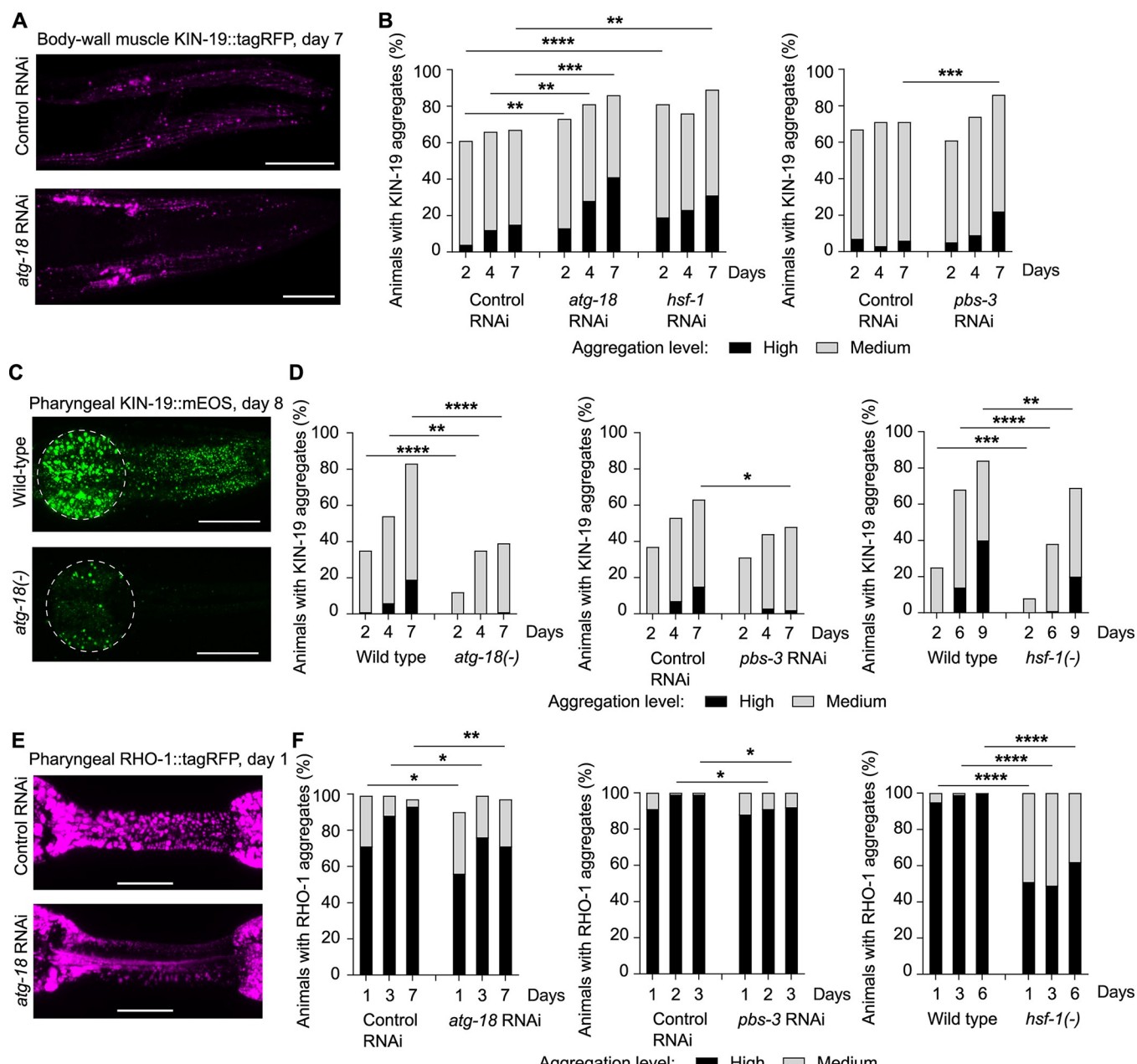

**Fig 1. PQC impairment leads to reduced protein aggregation in pharyngeal muscles.** (A, B) KIN-19 aggregation is increased in body-wall muscles upon PQC inhibition. Representative confocal images of KIN-19::tagRFP in the head body-wall muscles shown as maximum z-stack projections. Scale bar: 30 μm (A). Changes in KIN-19::tagRFP aggregation evaluated over time in the population with impaired macroautophagy (*atg-18*), reduced chaperone levels (*hsf-1*), and impaired proteasomal degradation (*pbs-3*) (B). (C, D) Animals during the course of aging with impaired PQC have less KIN-19::mEOS2 aggregates in the pharyngeal muscles. Representative confocal images of animals expressing KIN-19::mEOS2 in the pharynx shown as maximum z-stack projections with anterior pharyngeal bulb circled in white. Scale bar: 20 μm (C). Changes in KIN-19::mEOS2 aggregation evaluated over time in the population with impaired PQC (D). (E, F) Young animals with impaired PQC have less RHO-1::tagRFP aggregates in the pharyngeal muscles. Representative confocal images of RHO-1::tagRFP in the pharyngeal isthmus shown as maximum z-stack projections. Scale bar: 15 μm (E). Changes in RHO-1::tagRFP aggregation evaluated over time in the population with impaired PQC (F). *P*-values determined by Fisher's exact test, Chi-square test, and ordinal logistic regression. $*p < 0.05$, $**p < 0.01$, $***p < 0.001$, $****p < 0.0001$. See also S1 Data for number of animals evaluated and statistics and S1 and S2 Figs.

## Protection against pharyngeal protein aggregation in response to PQC impairment lessens proteotoxicity

We asked whether preventing protein aggregation during PQC failure confers a physiological advantage to the organism. Previously, evidence showed that age-dependent protein aggregates display amyloid-like structures, similar to pathological disease-associated amyloid aggregates and that this aggregation accelerates age-related functional decline [21]. Moreover, RHO-1 aggregates present in the pharyngeal muscles of young animals are sufficient to impair pharyngeal pumping, demonstrating that these aggregates are detrimental independent of aging processes. To detect changes in amyloid formation upon PQC impairment, we measured the fluorescence lifetime of fluorescent-label RHO-1. Amyloid structures within aggregates cause a characteristic drop in the fluorescence lifetime due to quenching [21,54]. Here, we found that impairment of PQC lead to lower levels of amyloid-like aggregates in the pharynx as demonstrated by a rescue of fluorescence lifetime compared to the control (Fig 2A). Next, we evaluated how PQC inhibition affects aggregate-related proteotoxicity. For this, we used HSF-1 inhibition with the hypomorphic *hsf-1* mutation, which leads to a strong reduction in RHO-1 aggregation in young animals (Fig 1F). Compared to inhibition of macroautophagy or proteasome activity, these mutants are relatively healthy [44]. Importantly, we found that *hsf-1* inhibition partially rescued the decline in pharyngeal pumping caused by RHO-1 aggregation (Fig 2B–2D). Thus, preventing protein aggregation in the pharynx in response to PQC failure appears to be a safety mechanism that benefits the organism at the physiological level.

## The safety mechanism displays substrate specificity

Next, we asked whether the safety mechanism is effective against RNA-binding proteins with prion-like domains. The stress granule marker and polyadenylate-binding protein 1 (PAB-1) forms stress-like granules and large immobile aggregates during aging in *C. elegans*. Unlike KIN-19 or RHO-1, PAB-1 contains a low complexity prion-like domain that promotes its liquid–liquid phase separation [22,55]. Previous work indicated that the aggregation of stress granule proteins and globular proteins is controlled in part by different mechanisms [22,55,56]. Notably, HSF-1 activity is crucial to prevent PAB-1 aggregation in the pharynx, raising the possibility that the safety mechanism does not target PAB-1 aggregation [22]. We confirmed this finding and show that PAB-1 aggregation in the pharynx is accelerated by inhibition of the proteasome and macroautophagy systems (S4 Fig). These findings imply that the safety mechanism displays a degree of specificity in targeting aggregating proteins.

## The safety mechanism prevents de novo formation of KIN-19 and RHO-1 aggregates

To gain insight into how protein aggregation is regulated in response to PQC failure, we investigated the dynamics of aggregate formation and removal. For this, we used KIN-19 and RHO-1 tagged with mEOS2, a green-to-red photoconvertible fluorescent protein [57]. By exposing the animals to intense blue light, all aggregates present are converted to emitting red fluorescence. Of note, the core region of some large aggregates was refractory to photoconversion (Fig 3 and as previously described [21]). Thereafter, we followed the rate of new (green emitting) aggregate formation and the rate of old (red emitting) aggregate removal over time. During aging, KIN-19 and RHO-1 are prone to form aggregates shortly after synthesis while old aggregates are slowly removed (Fig 3A–3C) [21]. Upon PQC impairment, the rate of old aggregate removal was either similar for KIN-19 aggregates or moderately enhanced for RHO-1 aggregates compared to control conditions (Fig 3A–3C). Therefore, disaggregation does not

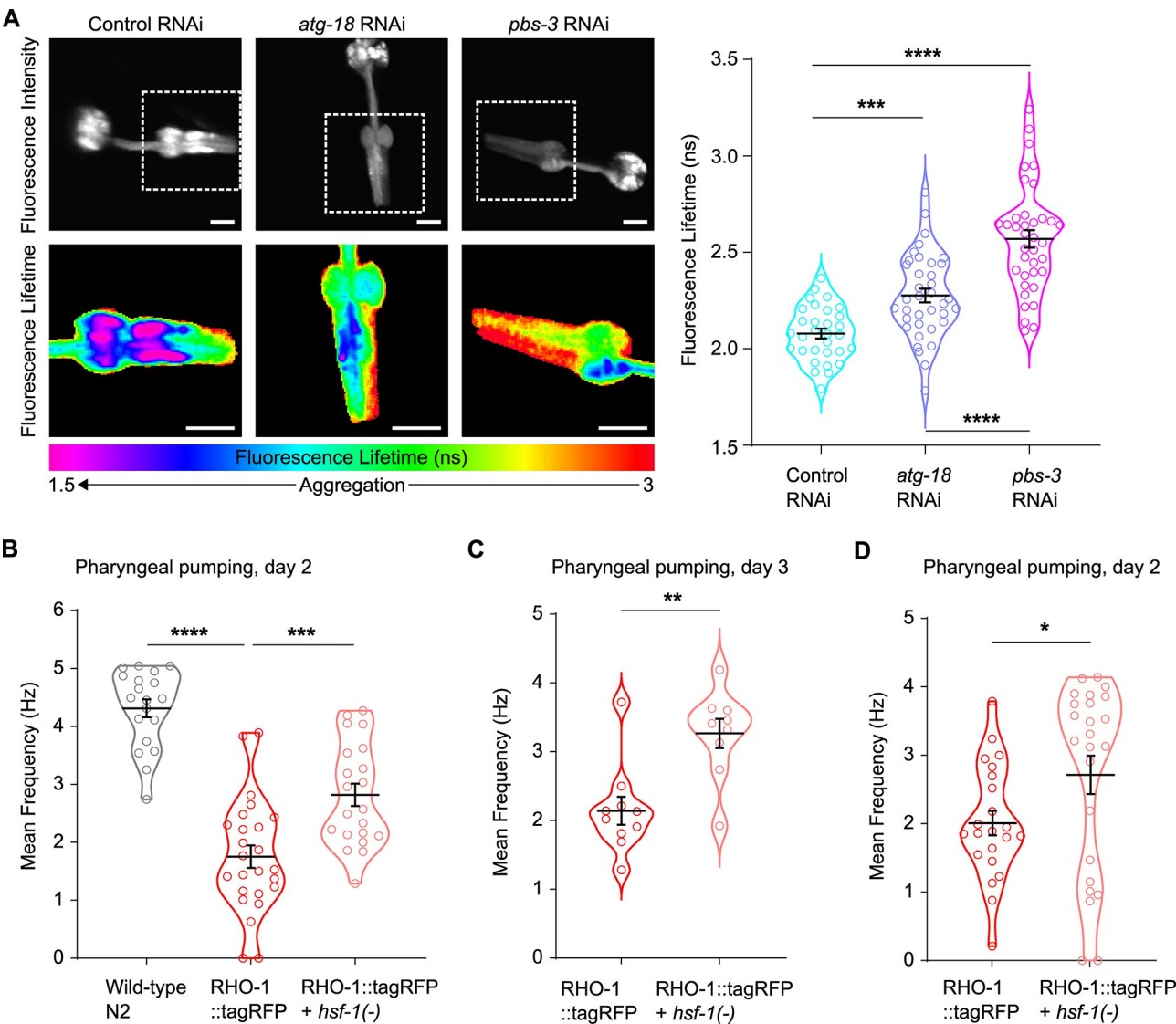

**Fig 2. Reducing RHO-1 aggregation alleviates amyloid formation and proteotoxicity.** (A) PQC failure leads to reduced formation of amyloid-like aggregates. Representative fluorescence images (left panel) and worm-averaged fluorescence lifetimes (right panel) of RHO-1::Venus in pharyngeal muscles of day 4 adults with macroautophagy impairment (*atg-18* RNAi) or proteasome impairment (*pbs-3* RNAi) compared to control conditions. Fluorescence intensity is displayed in upper left panel. Lower left panel shows zoomed-in intensity-weighted fluorescence lifetime images of the metacorpus and procorpus regions. Scale bar: 10 μm. In right panel, each dot represents averaged fluorescence lifetimes measured in individual animals compiled from 3 independent biological repeats. Bars show mean and SEM. (B–D) *hsf-1* mutants rescue pharyngeal pumping defects caused by RHO-1 aggregation. Each dot represents mean frequency measured in individual animals. Bars show mean and SEM. Three independent experiments are shown. *P*-values determined by one-way ANOVA with Holm–Sidak's multiple comparison (A), one-way ANOVA with Tukey's multiple comparisons test (B) and unpaired *T* test (C, D). *$p < 0.05$, **$p < 0.01$, ***$p < 0.001$, ****$p < 0.0001$. See also S1 Data for number of animals evaluated and statistics.

appear to play a major role in the safety mechanism. In contrast, we observed a striking effect on the formation of aggregates with newly synthesized protein. During proteasome or macro-autophagy disruption, the percentage of animals with green-emitting KIN-19 and RHO-1 aggregates gained over 48 h was strongly reduced compared to control conditions (Fig 3A–3C). Representative images show a delay in the recruitment of newly synthesized KIN-19 and RHO-1 tagged with mEOS2 (green emitting) to existing aggregates (red emitting) and a relative absence of de novo formed aggregates (Fig 3D and 3E). In the body-wall muscles, where

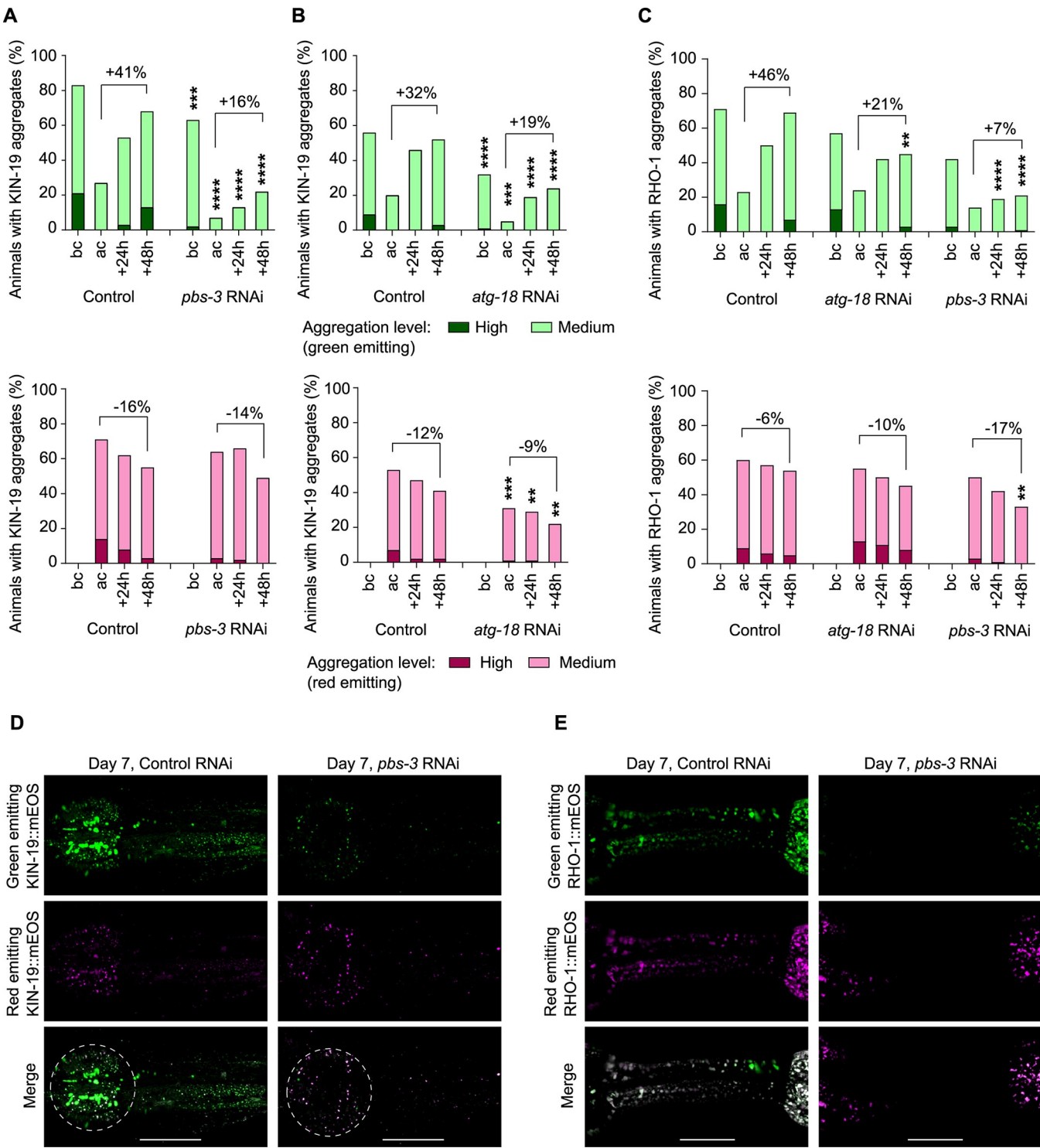

**Fig 3. De novo formation of aggregates in the pharynx is delayed in response to PQC failure.** (A, B) Formation of aggregates with green-labeled KIN-19::mEOS2 is delayed in the pharynx of animals with impaired protein degradation whereas the removal of old red-labeled aggregates is not affected. Proteasome inhibition by *pbs-3* RNAi (A) and macroautophagy inhibition by *atg-18* RNAi (B). (C) Formation of aggregates with green-labeled RHO-1::mEOS2 is delayed in the pharynx of animals with impaired protein degradation. (A–C) Photoconversion performed at day 5. Before conversion (bc), after conversion (ac). In graph, difference in percent of animals with aggregation between 48 h and after conversion. (D, E) Representative confocal images of day 7 animals, 48 h after photoconversion, expressing KIN-19::mEOS2 (D) or RHO-1::mEOS2 (E) in the pharynx shown as maximum z-stack projections with anterior pharyngeal bulb circled in white. Scale bar: 20 μm (D), 15 μm (E). *P*-values determined by Fisher's exact test comparing treatment versus control at the respective time point (A–C). **p < 0.01, ***p < 0.001, ****p < 0.0001. See also S1 Data for number of animals evaluated and statistics and S5 Fig.

we did not observe activation of a safety mechanism (Fig 1A and 1B), we found no difference between the rate of new aggregate formation and old aggregate removal when impairing proteasome activity compared to control conditions (S5A Fig). Together, these results reveal that the safety mechanism mainly limits the de novo formation and growth of aggregates in the pharynx.

## Newly synthesized proteins are removed before assembling into large aggregates

To understand how the safety mechanism interferes with the formation of new aggregates, we quantified changes in total levels of green-emitting KIN-19 upon PQC impairment in young animals during the early stages of protein aggregation. If the final assembly step into large visible aggregates is hindered, we would expect an accumulation of diffuse, newly synthesized proteins over time. In contrast, if aggregation is prevented by removing newly synthesized proteins, we would expect less green-emitting KIN-19 to accumulate over time. In support of the latter model, we observed a delay in the accumulation of newly synthesized KIN-19 over 48 h in response to proteasome and macroautophagy inhibition (S5B Fig). In control animals with mEOS2 protein alone, which does not aggregate [21], there was no reduction in fluorescent levels of newly synthesized green-emitting mEOS2 in response to PQC impairment (S5C Fig). Of note, overall higher levels of mEOS2 after 48 h is likely due to enhanced *kin-19* promoter activity with age as previously described [17]. We performed western blot analysis to measure total levels of KIN-19::mEOS and RHO-1::tagRFP in the pharynx. Quantification of the total pool of protein dissolved in sample buffer (including the newly synthesized unfolded and small aggregation species) did not show a clear consistent change upon proteasome inhibition (S6A Fig). In contrast, analysis of the sample buffer insoluble fraction containing highly aggregated species confirmed that animals with impaired proteasome degradation have lower aggregate levels (S6B Fig). Together with the acceleration of pharyngeal PAB-1 aggregation in response to PQC failure (S4 Fig), these results support the specificity of the safety mechanism and exclude a general decrease in translation. Altogether, our findings imply that the safety mechanism targets newly synthesized unfolded or small aggregation species for removal in order to avoid de novo formation of aggregates upon PQC disruption.

## Macroautophagy-independent lysosomal degradation protects against RHO-1 aggregation

Next, we examined how proteins are removed to avoid protein aggregation in response to PQC failure. As protein aggregation is reduced after inhibition of 2 major PQC systems responsible for protein degradation (proteasome and macroautophagy), this raises the possibility of an alternative removal pathway. We tested if the safety mechanism could be due to a compensatory up-regulation of one of the degradation systems [58,59]. However, combined inhibition of core PQC systems, for example, inhibition of both proteasome and macroautophagy-mediated degradation or inhibition of both HSF-1 and macroautophagy, failed to restore protein aggregation to wild-type levels (S7A and S7B Fig). In addition to macroautophagy, other forms of autophagy can target cytoplasmic material for lysosomal degradation, such as general microautophagy and chaperone-mediated autophagy [60–62]. Yet, it is unclear whether these types of autophagy occur in *C. elegans*. To investigate the potential role of the lysosome in degrading aggregating proteins without functional macroautophagy, we used loss-of-function mutants for the vacuolar proton translocating ATPase (VHA-12 subunit) responsible for lysosome acidification and the lysosomal membrane protein SCAV-3, a regulator of lysosome integrity [63]. Of note, SCAV-3 is prominently expressed in the pharynx. Both *vha-*

*12* and *scav-3* deletions impair lysosomal function leading to reduced levels and activity of lysosomal proteases [63,64]. We found that impairing lysosomal degradation through either *vha-12* or *scav-3* loss-of-function eliminated the safety mechanism triggered by macroautophagy inhibition and restored RHO-1 aggregation to levels seen in wild-type conditions without PQC disruption (Fig 4A and 4B). Thus, elimination of protein aggregation depends on macroautophagy-independent lysosomal degradation.

### The safety mechanism relies on components of the intracellular pathogen response

To investigate which factors may mediate the safety mechanism, we looked for genes selectively up-regulated when the safety mechanism is triggered, namely in response to PQC disruption together with overexpression in the pharynx of proteins prone to aggregate with age. For this, we performed RNA sequencing with whole animals to identify expression changes selectively induced by *hsf-1* or *atg-18* impairment in animals with pharyngeal KIN-19 and RHO-1 aggregates but not induced by PQC disruption in animals with body-wall muscle aggregates or in animals expressing the pharyngeal fluorescent tag alone (S8A Fig and S2 Data). With this experimental design, we identified 12 genes significantly up-regulated that could play a role in preventing protein aggregation in the pharynx (S1 Table and S2 Data). Among these, 2 genes have human orthologs: C01B10.4 has sequence similarities with Butyrylcholinesterase (E-value: 4e-30) and C53A5.11 has sequence similarities with Actin-binding protein IPP (E-value: 1.2e-27). Notably, 5 out of the 12 candidates are genes of unknown function previously identified as part of the host's intracellular pathogen response (IPR) (S2 Data) [65]. We targeted 8 of these 12 genes by RNAi. Upon PQC failure, knockdown of 6 out of 8 genes reproducibly contributed to restoring RHO-1 aggregation in the pharynx of *hsf-1* mutants (Fig 4C, S1 Table, and S3 Data), confirming their participation in the safety mechanism. In contrast in the wild-type background, the RNAi treatments had no effect on RHO-1 aggregation (S8B Fig and S3 Data), highlighting that PQC impairment is required to trigger the safety mechanism. Among the components discovered by RNA sequencing, PALS-5 is one of the most highly up-regulated genes in response to intracellular pathogens [65]. Overexpressed PALS-5 consistently colocalized with RHO-1 aggregates, suggesting an interaction between both proteins (Fig 4D). In contrast, we observed colocalization of PALS-5 with only a few PAB-1 aggregates (Fig 4E). Using a GFP transcriptional reporter for *pals-5*, we found expression of PALS-5 in the pharynx and intestine but not in the body-wall muscles (S8C–S8F Fig). Importantly, overexpression of PALS-5 in the pharynx prevented RHO-1 aggregation but not PAB-1 aggregation (Fig 4F–4H). Thus, PALS-5 on its own is sufficient to limit globular protein aggregation. Thus, these findings highlight the up-regulation of a select group of IPR factors, potentially in the pharynx, as an essential part of the safety mechanism.

### Discussion

Aberrant uncontrolled protein aggregation is detrimental to the organism as observed in neurodegenerative diseases and amyloidosis as well as during aging [8,21]. Enhancing proteostasis is a promising strategy to counteract proteotoxicity and considered in efforts to develop an effective therapy for diseases of protein aggregation [66]. In the current study, we used intrinsically aggregation-prone proteins in *C. elegans* to understand the proteostasis of age-dependent protein aggregation. We describe a novel safety mechanism triggered to ensure tissue-specific resistance to protein aggregation and its associated proteotoxicity in response to severe and systemic PQC impairment. We name this mechanism: safeguard against protein aggregation (SAPA).

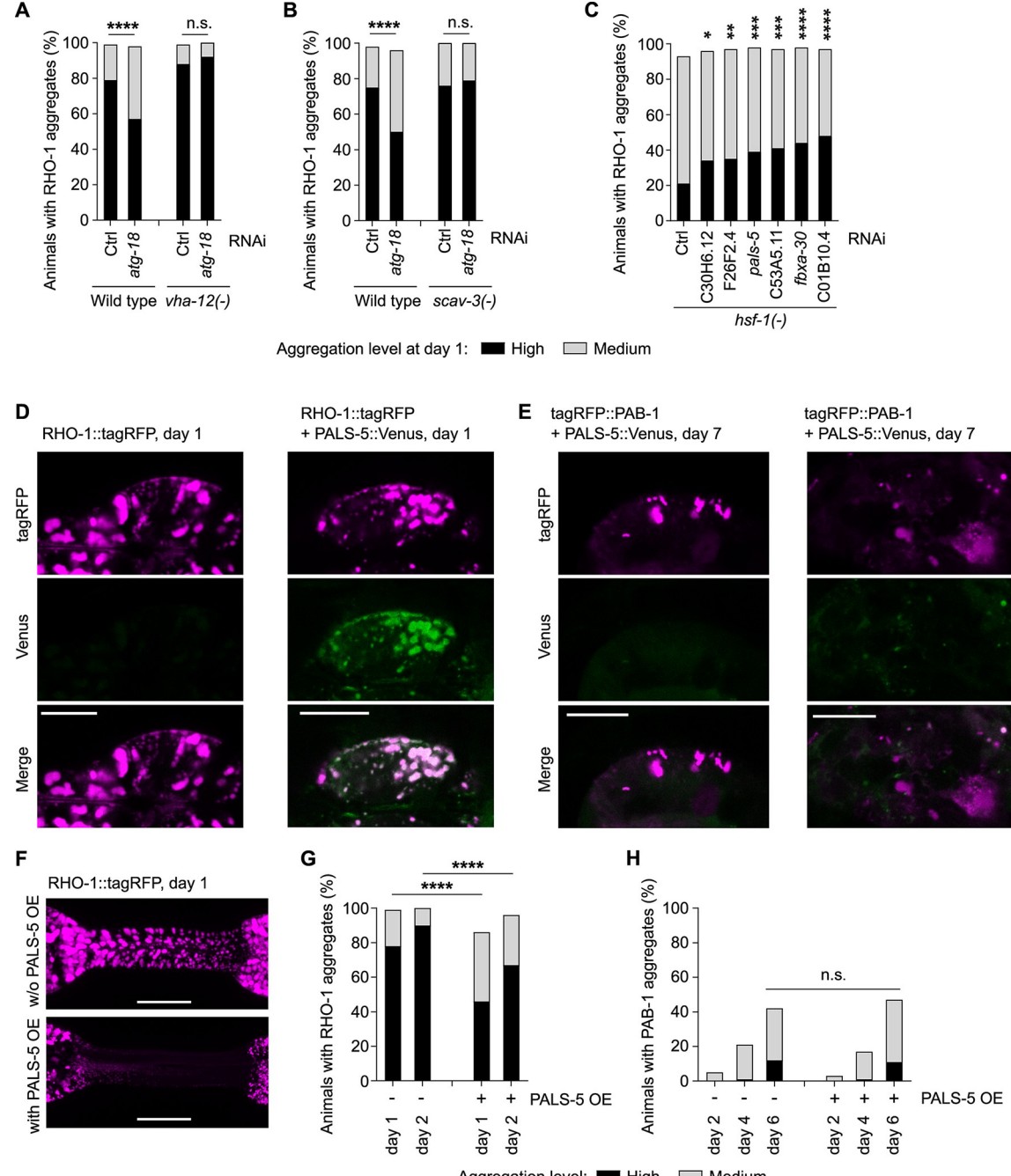

**Aggregation level at day 1:** ■ High ☐ Medium

**Aggregation level:** ■ High ☐ Medium

**Fig 4. The safety mechanism relies on macroautophagy-independent lysosomal degradation and factors involved in the intracellular pathogen response.** (A, B) Reduced pharyngeal RHO-1 aggregation upon macroautophagy impairment (*atg-18* RNAi) is restored by impairing lysosomal degradation (*vha-12(-)* (A) and (*scav-3(-)* (B)). Changes in RHO-1::tagRFP aggregation at day 1 in the worm population. (C) Knockdown by RNAi of genes identified by RNA-seq as selectively up-regulated by PQC failure in transgenic animals with pharyngeal RHO-1 and KIN-19 aggregation. Quantification of RHO-1::tagRFP aggregation at day 1 with PQC failure induced by *hsf-(-)*. (D, E) Representative confocal single plan images of pharyngeal RHO-1::tagRFP (magenta, 0.5% laser), PALS-5:: Venus (green, 5% laser), tagRFP::PAB-1 (magenta, 10% laser), scale bar 10 μm. PALS-5 colocalizes with RHO-1 aggregates (right panel) compared to no signal in the absence of PALS-5 OE (left panel) (D). No co-localization (left panel) or partial co-localization of PALS-5 with a few PAB-1 aggregates (right panel) (E). (F, G) PALS-5 overexpression delays RHO-1 aggregation in the pharynx. Representative confocal images of RHO-1::tagRFP in the pharyngeal isthmus in animals without (w/o) PALS-5 overexpression (OE) (top) and with PALS-5 OE (below), shown as maximum z-stack projections. Scale bar: 15 μm (F). Quantification of RHO-1::tagRFP aggregation in the worm population (G). (H) PALS-5 overexpression does not influence PAB-1 aggregation in the worm population. *P*-values determined by Fisher's exact test (A, B), ordinal logistic regression analysis comparing all RNAi treatments to the control

RNAi (C) and Fisher's exact test and Chi-square test comparing same days with and without PALS-5 OE (G, H). $^*p < 0.05$, $^{**}p < 0.01$, $^{***}p < 0.001$, $^{****}p < 0.0001$. See also S1 Data for number of animals evaluated and statistics, S2 Data for RNA-seq data, S3 Data for repeats, S8 Fig.

Our findings reveal that the safety mechanism does not target all aggregating substrates equally. SAPA was highly effective in preventing the aggregation of 2 normally globular proteins, casein kinase I isoform alpha (KIN-19) and ras-like GTP-binding protein rhoA (RHO-1). In contrast, SAPA was not effective against the aggregation of the stress granule marker, PAB-1. Previous findings reveal that interventions such as reduced insulin/IGF-1 signaling, mild chronic stress and treatment with lipoic acid strongly modulate the aggregation of RNA-binding proteins capable of liquid–liquid phase separation but have limited effect on the aggregation of globular proteins [22,55,56]. Whereas the aggregation of RHO-1 and KIN-19 starts shortly after synthesis, PAB-1 aggregation likely occurs in the context of aberrant stress granule formations [22,55]. Thus, the detected selectivity of SAPA may be explained by differences in the aggregation process of non-stress granule components compared to stress granule components.

We found that SAPA is triggered in response to severe systemic perturbations of key systems maintaining protein homeostasis, namely protein folding by chaperones and protein degradation by the proteasome and macroautophagy. SAPA is likely distinct from previously identified compensatory mechanisms activated in response to PQC impairment. Particularly well characterized is the up-regulation of macroautophagy in response to proteasome defects to ensure the removal of excessive levels of ubiquitinated proteins [58,59]. Yet, we found that SAPA is still functional when inhibiting both macroautophagy and proteasome-mediated degradation (S7 Fig). In response to macroautophagy impairment, Golgi membrane-associated degradation (GOMED) is induced to compensate [67]. GOMED is suppressed by Ulk-1 deletion (homolog of *C. elegans unc-51*). In contrast, SAPA is triggered by impaired *unc-51* function. Therefore, SAPA is likely also distinct from GOMED. Removal of protein aggregates by expulsion from neurons through exopher formation is enhanced in *C. elegans* during proteasome, macroautophagy, or *hsf-1* inhibition [39]. Yet, it is unlikely that exopher formation greatly contributes to protecting pharyngeal muscle cells from age-dependent protein aggregation as the pharynx is isolated from the pseudocoelom by a thick basal lamina. Instead, our data reveal that in response to core PQC failure, newly synthesized proteins are cleared before forming large aggregates by noncanonical degradation. While macroautophagy is the best characterized form of autophagy, cytosolic proteins can be targeted to the lysosome through other forms of autophagy. Chaperone-mediated autophagy specifically targets proteins for import into the lysosome through HSC70 and LAMP2A [68]. However, no ortholog of LAMP2A has been identified in *C. elegans* [69]. Another possibility to degrade cytosolic proteins without using autophagosomes is general microautophagy where the cargo is taken up by vesicles formed at the surface of the lysosome or by late endosomes [60–62]. Microautophagy has yet to be characterized in *C. elegans* [69], yet, 2 recent studies provide evidence for macroautophagy-independent degradation of cytosolic cargo by the lysosome evocative of microautophagy [70–72]. Further work is needed to establish how SAPA would rely on microautophagy to stop the aggregation of certain globular proteins during core PQC failure.

Selective vulnerability to protein aggregation remains a conundrum. Underlying genetic risk factors involved in protein degradation have been associated with major neurodegeneration disorders such as Alzheimer's disease, Parkinson's disease, and amyotrophic lateral sclerosis [25]. Moreover, increasing evidence highlights the importance of lysosomes in diseases of protein aggregation [73]. Yet, differences in aggregating protein turnover between cell types is

probably determined by tissue-specific secondary proteostasis regulators [74] rather than differences in the core components of the degradative systems, which are ubiquitously expressed. In *C. elegans*, our analysis shows that the tissue-specific effect of SAPA is likely related to the expression of genes enriched in uncharacterized IPR factors. Responses to proteostress are often systemically coordinated by inter-tissue signaling [75]. Yet, the involvement of IPR components, which would normally be expressed in the digestive system directly exposed to pathogens, points to a cell autonomous response. Overexpression in the pharynx of PALS-5, a commonly used reporter for IPR with unknown function, was sufficient to limit protein aggregation. Moreover, human butyrylcholinesterase, the potential ortholog of SAPA component C01B10.4, inhibits amyloid-β aggregation in vitro [76]. Protein–protein interaction domains such as F-box and MATH (meprin-associated Traf homology) are overrepresented in IPR genes and we found 3 F-box domains and 1 MATH domain coding genes among the SAPA components. Interestingly, a number of F-box proteins act as adapters in ubiquitin-mediated protein degradation [77] and 2 F-box proteins were recently shown to mediate thermotolerance in the digestive tract [78]. Thus, an intriguing possibility is that the SAPA components identified are anti-aggregation factors selectively targeting unfolded proteins or small intermediate aggregating species for degradation by lysosomes localized in the pharynx.

The clearance rate of disease-associated aggregating proteins is correlated with neuronal survival [79]. Similarly, we found that the safety mechanism triggered by PQC failure alleviated proteotoxicity and partially restored the pharyngeal pumping activity in *C. elegans*. Strikingly, SAPA was so effective that animals during the mid-life aging process were protected against protein aggregation even compared to animals of similar ages, not exposed to PQC failure. Why does the organism choose to prioritize resources to protect foremost the pharynx rather than body-wall muscles during PQC failure? The reason could be the fundamental role of the pharynx in feeding and its direct exposure to natural pathogens. The pharynx is a contractile organ that pumps bacterial food from the mouth of the worm into its intestine. By grinding up bacteria, the pharynx limits bacterial colonialization and infection in the intestine. Importantly, bacterial infection in the pharynx is associated with early death [80]. The digestive tract is also the point of entry for obligate intracellular pathogens such as viruses. Upon infection, the host triggers the IPR transcriptional program, which is distinct from the response to extracellular pathogens [65,81,82]. One of the host's defense strategies against intracellular and extracellular pathogens is to improve resilience by enhancing proteostasis [2,65,81,83]. Our work reveals that upon failure of PQC the organism co-opts part of the response to intracellular pathogens by inducing IPR genes such as *pals-5* to protect the pharynx against excessive proteostress. Consistent with this, *pals-5* expression in the digestive tract is induced by heat stress and proteasome inhibition as well as by intracellular pathogens [65,81]. Whether these select IPR genes used by SAPA protect also against endogenous protein aggregation during an infection remains to be determined.

In conclusion, our findings characterize a novel safety mechanism acting in a specific tissue to suppress age-dependent protein aggregation and its proteotoxicity in response to failure of core PQC systems. Such mechanisms could be key modulators involved in defining the differences in aggregate composition between cells and tissues with age. A better understanding of selective targeting of aggregating proteins to the lysosome should help to design strategies to restore proteome stability in vulnerable aged tissues.

## Materials and methods

### Strains

Wild type: N2

Transgenics:

CF3166: *muEx473[pkin-19::kin-19::tagrfp + Ptph-1::gfp]*

CF3317: N2; *muEx512[Pkin-19::tagrfp + Ptph-1::GFP]*

CF3649: N2; *muIs209[Pmyo-3::kin-19::tagrfp + Ptph-1::gfp]*

CF3706: N2; *muEx587[Pkin-19::kin-19::meos2 + Punc-122::gfp]*

DCD13: N2; *uqIs9[Pmyo-2::rho-1::tagrfp + Ptph-1::gfp]*

MAH215: *sqIs11[Plgg-1::mCherry::GFP::lgg-1 + rol-6]*

DCD45: N2B; *muIs115[Phsf-1::hsf-1 cDNA + Pmyo-3::GFP]; uqIS10[Pkin-19::kin-19::mEOS2 + Punc-122::gfp]*

DCD69: N2; *uqEx4[Pmyo-3::kin-19::meos2]*

DCD83: *ttTi5605II; unc-119(ed3)III; uqEx11[Pmyo-2::rho-1::meos2 + Punc-122::gfp + cb-unc-119(+)]*

DCD88: N2; *muIs115[Phsf-1::hsf-1 cDNA + Pmyo-3::GFP]; uqIs9[Pmyo2::rho1::tagrfp + Ptph-1::gfp]*

DCD92: *hsf-1(sy441)I; uqIs9[Pmyo-2::rho1::tagrfp + Ptph-1::gfp]*

DCD146: N2; *uqIs12[Pmyo-2::rho-1::venus]*

DCD173: *hsf-1(sy441)I; muEx587[Pkin-19::kin-19::meos2 + Punc-122::gfp]*

DCD174: *atg-18(gk378)V; muEx587[Pkin-19::kin-19::meos2 + Punc-122::gfp]*

DCD190: N2B; *uqIs9[Pmyo2::rho1::tagrfp + Ptph-1::gfp]; uthEx557 [Psur-5::rpn-6 + Pmyo-3::GFP]*

DCD214: N2; *uqIs24[Pmyo-2::tagrfp::pab1]*

DCD245: N2; *uqEx49[Pkin-19::meos2]*

DCD249: *hsf-1(sy441)I; uqIs24[Pmyo-2::tagrfp::pab1]*

DCD258: *hsf-1(sy441)I; muIs209[Pmyo-3::kin-19::tagrfp + Ptph-1::gfp]*

DCD324: *unc-51(e369)V; uqIs9[Pmyo-2::rho1::tagrfp + Ptph-1::gfp]*

DCD326: *vha-12(ok821)X; uqIs9[Pmyo-2::rho1::tagrfp + Ptph-1::gfp]*

DCD340: *unc-51(e369)V; uqIs24[Pmyo-2::tagrfp::pab1]*

DCD345: *scav-3(qx193)III; uqIs9[Pmyo-2::rho1::tagrfp + Ptph-1::gfp]*

ERT54: *jyIs8 [Ppals-5::GFP + Pmyo-2::mCherry] X*

DCD411: N2; *uqIs9[Pmyo2::rho1::tagrfp + Ptph-1::gfp]; uqEx64[Pkin-19::pals-5::mVenus::histag + punc-122::gfp]*

DCD415: N2; *uqIs24[Pmyo-2::tagrfp::pab1]; uqEx64[Pkin-19::pals-5::mVenus::histag + Punc-122::gfp]*

XW8056: N2; *qxIs430[Pscav-3::scav-3::gfp]*

## Cloning and strain generation

Cloning and strain generation were performed as previously described [21]. Construct *pkin-19::pals-5::mVenus::histag* and coinjection marker *punc-122::gfp* were injected at 50 ng μl$^{-1}$ each. Genetic crosses were made to transfer transgenes to the appropriate genetic background. The presence of the mutant allele was verified by polymerase chain reaction (PCR).

## *C. elegans* maintenance and RNAi knockdown

All strains were kept at 15°C on NGM plates (51 mM NaCl, 2% w/v Difco Bacto Agar, 0.25% w/v Bacto Peptone, 1 mM MgSO$_4$, 1 mM CaCl$_2$, 0.0005% w/v Cholesterol, 25 mM Potassium Phosphate buffer) inoculated with OP50 using standard techniques. Age-synchronization was achieved by transferring adults of the desired strain to 20°C and selecting their progeny at L4 stage. Adult mutants with *atg-18(-)* and *unc-51(-)* were kept at 15°C for laying eggs. From L4 stage, all experiments were performed at 20°C. RNAi treatment was performed by feeding as

previously published [84]. The RNAi clones were acquired from the Marc Vidal or the Julie Ahringer RNAi feeding library (Source BioScience, United Kingdom) and sequenced. HT115 containing the empty vector L4440 was used as control. To start RNAi treatment from egg stage, adults were allowed to lay eggs on bacterial lawn expressing the dsRNA. To avoid developmental defects, proteasome subunits and *hsp-1* were targeted by RNAi from L4 stage. Day 1 of adulthood starts 24 h after L4. To enhance the RNAi effects, F2 generation from animals subjected to RNAi treatment were evaluated when indicated.

### Treatment with MG-132

The animals were exposed to MG-132 (J63250, Thermo Fisher) in liquid culture in a 96-well plate starting from larval stage L4 in a total volume of 50 μl S-Complete per well (100 mM NaCl, 50 mM Potassium phosphate (pH 6), 10 mM potassium citrate, 3 mM MgSO$_4$, 3 mM CaCl$_2$, 5 μg/ml cholesterol, 50 μM ethylenediaminetetraacetic acid (EDTA), 25 μM FeSO$_4$, 10 μM MnCl$_2$, 10 μM ZnSO$_4$, 1 μM CuSO$_4$) supplemented with heat-killed OP50 and 50 μg/ml carbenicillin. Per experiment, 7 to 11 wells with 13 animals each were treated with 0.5 mM MG-132 or an equivalent volume of DMSO. Aggregation quantification was performed at day 2 of adulthood.

### Imaging

For confocal analysis using a Leica SP8 confocal microscope or a Leica Stellaris 8 with the HC PL APO CS2 63×/1.30 NA glycerol objective, worms were mounted onto slides with 2% agarose pads using 2 μM levamisole for anesthesia. For body-wall muscle images, worms were fixed in 4% paraformaldehyde (PFA) for 10 min at room temperature and imaged with the HC PL APO CS2 63×/1.40 NA oil objective. Leica HyD hybrid detector was used to detect KIN-19::mEOS2 (excitation: 506 nm, emission: 508 to 550 nm), PALS-5::Venus (excitation: 515 nm, emission: 521 to 545 nm), tagRFP::PAB-1 (excitation: 555 nm, emission: 565 to 650 nm). Leica PMT detector was used to detect RHO-1::tagRFP (excitation: 555 nm, emission: 565 to 620 nm).

### Western blot analysis

DCD146 and CF3706 transgenics were selected at L4 stage and grown on either the control empty vector (L4440) or *pbs-3* RNAi agar plates at 20°C. Per condition, 100 animals were lysed in NuPage LDS sample buffer (Invitrogen) supplemented with 4 M urea and NuPage sample reducing agent (Invitrogen) on days 4 and 7 of adulthood, respectively. Lysates were snap frozen in liquid nitrogen, thawed, heated at 70°C for 10 min and frozen at –80°C overnight. After centrifugation at 18,400 g for 10 min, supernatants were collected as the total protein sample. The pellet from centrifugation was resuspended in 8 M urea and reprocessed as above. After separation by SDS-PAGE and transfer, western blot analysis was performed with anti-Rho1 (1:4,000, gift from Asako Sugimoto), anti-CK1 (1:500, 2655, Cell Signaling Technology), anti-actin (1:500, MAB1501, Sigma-Aldrich), anti-ubiquitinylated proteins (1:500, FK2 04–263, Sigma-Aldrich) and detected using horseradish peroxidase (HRP)-conjugated anti-rabbit (1:2,000, 7074S, Cell Signaling Technology) and anti-mouse secondary antibodies (1:2,000, 7076S, Cell Signaling Technology). Blots were quantified by ImageJ and normalized to actin levels.

### Photoconversion of mEOS2-tag and quantification of fluorescence levels

Photoconversion was performed as previously described by illuminating worms placed on a plate with blue fluorescence (387/11 BrightLine HC, diameter 40 mm) [21]. For all

conversions, transgenic animals were exposed to blue light 4 times for 5 min, with 2 min pauses between exposures except for transgenics expressing *pmyo-2*::*RHO-1*::*mEOS2*, which were exposed to blue light 5 times for 6 min, with 2 min pauses. For quantification of fluorescence levels, worms were mounted onto slides with 2% agarose pads using 2 μM levamisole for anesthesia. Using an Axio Observer Z1 (Zeiss), levels of green fluorescence (eGFP set 38HE, excitation 470 ± 40 nm, emission 525 ± 50 nm) were detected. Quantification of fluorescence levels was determined using ImageJ with the oval-shaped selection tool placed around the anterior bulb. Total fluorescence was obtained by subtracting the mean intensity in the anterior bulb from the mean background intensity and multiplying by the area measured.

## Aggregation quantification in vivo

Aggregation levels were determined using Leica fluorescence microscope M165 FC with a Planapo 2.0× objective. Aggregation was quantified following pre-set criteria adapted to the transgene expression pattern and levels in the different transgenic *C. elegans* models: Animals expressing *Pkin-19*::*KIN-19*::*mEOS2* or *Pkin-19*::*KIN-19*::*TagRFP* were divided into less than 10 puncta (low aggregation), between 10 and 100 puncta (medium aggregation), and over 100 puncta in the anterior pharyngeal bulb (high aggregation) [17,21]. Because of extensive RHO-1 aggregation in young animals overexpressing *Pmyo-2*::*RHO-1*::*TagRFP* or *Pmyo-2*::*RHO-1*::*mEOS2*, aggregation was only quantified in the isthmus: animals with no aggregation (low aggregation), animals with aggregation in up to 50% (medium aggregation), and animals with aggregation in more than 50% (high aggregation) of the isthmus. Aggregation of RHO-1::Venus in animals overexpressing *Pmyo-2*::*RHO-1*::*Venus* was evaluated in the anterior bulb and procorpus. Animals were divided into animals with aggregation in up to 50% in procorpus (low aggregation), more than 50% in procorpus (medium aggregation), and over 10 puncta in the anterior pharyngeal bulb (high aggregation). Animals overexpressing *Pmyo-2*::*tagRFP*::*PAB-1* were divided into less than 10 puncta (low aggregation) and over 10 puncta (medium aggregation) in the posterior bulb and over 10 puncta in the anterior bulb (high aggregation) [22]. Animals overexpressing *Pmyo-3*::*KIN-19*::*TagRFP* were divided into over 15 puncta in the head or the middle body region (low aggregation), over 15 puncta in the head and the middle body region (medium aggregation), and over 15 puncta in head, middle body, and tail region (high aggregation). The same categories defined for animals overexpressing *Pmyo-3*::*KIN-19*::*TagRFP* were used to evaluate animals overexpressing *Pmyo-3*::*KIN-19*::*mEOS2* with a cutoff of 10 puncta instead of 15 to account for slightly lower aggregation levels. Counting was done in a blind fashion for all conditions.

## TCSPC-FLIM

For fluorescent lifetime imaging microscopy analysis, worms were mounted onto slides with 4% agarose pads using 2 μM levamisole for anesthesia. Samples were imaged on a home-built confocal fluorescence microscope equipped with a time-correlated single photon counting (TCSPC) module. A pulsed, supercontinuum laser (Fianium Whitelase, NKT Photonics) provided excitation a repetition rate 40 MHz. This was passed into a commercial microscope frame (IX83, Olympus) through a 40× oil objective (EVOS UPlanFLN 40ox, 1.3 NA, Olympus). The excitation and emission beams were filtered through bandpass filters centered at 510 and 542 nm, respectively (FF03-510/20 and FF01-542/2, Semrock). Laser scanning was performed using a galvanometric mirror system (Quadscanner, Aberrior). Emission photons were collected on a photon multiplier tube (PMT, PMC150, B&H GmBH) and relayed to a time-correlated single photon counting card (SPC830, B&H GmBH). Images were acquired at 256 × 256 pixels for 120 s (i.e., 10 cycles of 12 s). Photon counts were kept below 1% of laser

emission photon (i.e., SYNC) rates to prevent photon pile-up. TCSPC images were analyzed using an in-house phasor plot analysis script (https://github.com/LAG-MNG-CambridgeUniversity/TCSPCPhasor), from which fluorescence lifetime maps and phasor plots were generated. Fluorescence lifetimes of the metacorpus and procorpus regions for each imaged worm were calculated. FLIM images for illustrative purposes have been rendered with FLIMFIT 5.1.1 [85] and the color map has been adjusted using Python 3.8.16.

## Quantification of autophagosome/autolysosome ratios

Animals expressing *Plgg-1*::*mCherry*::*GFP*::*lgg-1* were treated with *atg-18* RNAi or control L4440 RNAi at L4 stage and analyzed at day 1 of adulthood. Imaging was performed using an Olympus SpinSR spinning disk confocal microscope using an Olympus UPLAPO60XOHR 60× oil immersion objective (GFP: 488/525 ± 50 nm; mCherry: 561/595 ± 31 nm). Worms were mounted onto microscope slides on 2% agarose pads in using 2 μM levamisole. Autophagosome/autolysosome ratios were calculated in the posterior pharyngeal bulb as described in previous study [30].

## Animal collection for RNA sequencing

Adults were allowed to lay eggs during 7 h at 20˚C on bacterial lawn expressing dsRNA for *atg-18*, *hsf-1*, and empty vector L4440. To synchronize the animals, progeny at L4 stage were selected and kept at 20˚C until collection. Animals overexpressing *Pmyo-2*::*RHO-1*::*TagRFP* (DCD13) were collected at day 2, 300 worms per condition. Animals overexpressing *Pkin-19*:: *KIN-19*::*TagRFP* (CF3166) were collected at day 7, 230 worms per condition. Control animals overexpressing *Pmyo-3*::*KIN-19*::*TagRFP* (CF3649) were collected at day 4, 300 worms per condition. Control animals overexpressing *Pkin-19*::*TagRFP* (CF3317) were collected at day 7, 300 worms per condition. Collected animals were washed twice in M9 and snap frozen.

## RNA sequencing

Total RNA was isolated using Direct-zol RNA Mini-Prep Kit following the manufacturers' instructions (Zymo Research, California, United States of America). RNA concentration was quantified using Qubit (Invitrogen Life Technologies, California, USA) and Nanodrop (PEQ-LAB Biotechnologie GmbH, Erlangen, Germany) measurements. RNAseq libraries were prepared using TruSeq RNA library preparation kit v2 (Illumina, California, USA) according to the manufacturer's instructions from 1 μg of total RNA in each sample. Libraries were quantified using Qubit and Bioanalyzer measurements (Agilent Technologies, California, USA) and normalized to 2.5 nM. Samples were sequenced as 150 bp paired end reads on multiplexed lanes of an Illumina HiSeq3000 (Illumina, California, USA). All sequencing data has been submitted to the European Nucleotide archive under the study accession PRJEB41493.

## Analysis of RNA-seq data

Raw RNA-seq reads were aligned to the *C. elegans* reference genome (Wormbase release WS250) with the help of tophat2 (version 2.0.14, default options) [86]. We used *C. elegans* gene annotations (Wormbase release WS250) to quantify expression levels and to perform tests for differential expression for all pairwise comparisons of samples with the software cuff-diff (version 2.2.1, default options) [87]. For further analysis, any gene with a $P$-value $<0.05$ was considered as candidate for differential expression.

## Pharyngeal pumping analysis

Electrical activity of the pharyngeal pumping was measured using the NemaMetrix Screen-Chip System (NemaMetrix, Eugene, Oregon, USA) as previously described [21].

## Statistical analysis

For analysis of aggregation quantification in vivo, two-tailed Fisher's exact test and Chi-square test were performed with GraphPad. Fisher's exact test was used for comparisons with 2 categories of aggregate levels where 2 of the 3 categories were combined together when a category counted less than 5 animals. A one-way ANOVA test with Holm–Sidak's multiple comparison was performed using GraphPad to analyze differences in fluorescence lifetimes. For analysis of changes in fluorescence intensities, unpaired $t$ test with two-tail distribution or a one-way ANOVA with Tukey's multiple comparison test were performed with GraphPad. To analyze the effect on protein aggregation of multiple RNAi treatments compare to control RNAi, we used an ordinal logistic regression model, which was performed using R and its MASS package. Enrichment analysis of safety mechanism related components in Data File 2 was performed with WormExp (http://wormexp.zoologie.uni-kiel.de/wormexp/), category Microbes, one-sided Fisher's exact test with Bonferroni correction $P < 0.05$.

## Supporting information

**S1 Fig. Multiple interventions targeting PQC result in a tissue-specific reduction of protein aggregation.** (A, B) Animals between day 8 and day 10 of adulthood with impaired PQC have less KIN-19 aggregates in the pharynx. Representative confocal images of animals expressing KIN-19::mEOS2 in the pharynx shown as maximum z-stack projections with anterior pharyngeal bulb circled in white. Scale bar: 20 μm (A). Changes in KIN-19::mEOS2 aggregation (left panel) or in KIN-19::tagRFP aggregation (middle and right panels) evaluated over time in the population with impaired macroautophagy (*atg-18* RNAi), impaired proteasomal degradation (*rpt-6* RNAi), and reduced chaperone levels (*hsf-1* RNAi) (B). (C–E) Young animals with impaired PQC have less RHO-1 aggregates in the pharynx. Representative confocal images of RHO-1::tagRFP in the pharyngeal isthmus shown as maximum z-stack projections. Scale bar: 15 μm (C). Changes in RHO-1::tagRFP aggregation evaluated in the population with impaired macroautophagy (*unc-51(-)*) (D). Chemical inhibition of proteasome activity with 0.5 mM MG-132 in 3 independent experiments (E). (F) Changes in RHO-1::Venus aggregation (left panel) or in KIN-19::mEOS aggregation (right panel) evaluated over time in the population with reduced chaperone *hsp-1* levels (*hsp-1* RNAi). (G) Representative confocal images of animals expressing RHO-1::Venus and KIN-19::mEOS in the pharynx shown as maximum z-stack projections with anterior pharyngeal bulb circled in white. Scale bar: 20 μm (B, D-F) *P*-values determined by Fisher's exact test and Chi-square test. \*\*$p < 0.01$, \*\*\*$p < 0.001$, \*\*\*\*$p < 0.0001$. See also S1 Data for number of animals evaluated and statistics. (EPS)

**S2 Fig. PQC is effectively impaired by RNAi treatment.** (A) GFP fluorescence in the anterior pharyngeal bulb is effectively reduced by *gfp* RNAi in SCAV-3::GFP overexpressing day 1 animals. (B) Increased autophagosome/autolysosome ratios measured in the pharyngeal posterior bulb of *atg-18* RNAi treated day 1 animals with mCherry::GFP::LGG-1. Dots show data of individual animals and bars show mean and SEM. *P*-values determined by unpaired *T* test. \*\*\*\*$p < 0.0001$. (A, B). (C) Ubiquitinated proteins accumulate upon *pbs-3* RNAi treatment. Immunoblots with 2 independent replicates with lysates dissolved in sample buffer at day 7 of adulthood are shown with histogram quantifications normalized to actin levels. See also S1

Data for number of animals evaluated and statistics.
(EPS)

**S3 Fig. Enhancing PQC delays protein aggregation in pharyngeal muscles.** (A) Increased chaperone levels by HSF-1 overexpression (OE) or increased proteasome activity by RPN-6 overexpression delays RHO-1::tagRFP aggregation with age. (B) Increased chaperone levels by overexpression of HSF-1 (OE) delays KIN-19::mEOS2 aggregation with age. $P$-values determined by Fisher's exact test and Chi-square test. $^*p < 0.05$, $^{***}p < 0.001$, $^{****}p < 0.0001$. See also S1 Data for number of animals evaluated and statistics.
(EPS)

**S4 Fig. The safety mechanism does not target stress granule PAB-1 protein aggregation.** Inhibition of macroautophagy (left panel, *unc-51(-)*), proteasome activity (middle panel, *pbs-3* RNAi), or chaperone levels (right panel, *hsf-1(-)*) accelerate tagRFP::PAB-1 aggregation in pharyngeal muscles with age. $P$-values determined by Fisher's exact test and Chi-square test. $^{****}p < 0.0001$. See also S1 Data for number of animals evaluated and statistics.
(EPS)

**S5 Fig. Formation of aggregates in pharynx is avoided by promoting removal of newly synthesized protein.** (A) The rate of new green-labeled KIN-19::mEOS2 aggregate formation in the body-wall muscles is similar in control and proteasome disruption conditions (*pbs-3* RNAi). Conversion was done at day 2. Before conversion (bc), after conversion (ac). In graph, difference in percent of animals with aggregation between 48 h and after conversion. (B) After photoconversion of pharyngeal KIN-19::mEOS2 in young day 2 adults, monitoring of total green fluorescence reveals reduced accumulation over time of newly synthesized protein upon proteasome impairment (left panel) or macroautophagy impairment (right panel). Relative fluorescence represents total fluorescence of treatment normalized to control RNAi after conversion. (C) After photoconversion of pharyngeal mEOS2 alone in young day 2 adults, monitoring of total green fluorescence shows accumulation of newly synthesized fluorescent tag in both control treatment and upon proteasome impairment (left panel, *pbs-3* RNAi) or macroautophagy impairment (right panel, *atg-18* RNAi). Relative fluorescence represents total fluorescence of treatment normalized to control RNAi after conversion. Each dot represents fluorescence measured in individual animals. Bars show mean and SEM. $P$-values determined by one-way ANOVA with Tukey's multiple comparison test (B, C). $^{**}p < 0.01$, $^{****}p < 0.0001$. See also S1 Data for number of animals evaluated and statistics.
(EPS)

**S6 Fig. Quantification of total RHO-1 and KIN-19 levels in pharynx.** (A) Immunoblots with 2 independent replicates with lysates dissolved in sample buffer are shown with histogram quantifications of KIN-19::mEOS2 (day 7, left panel) and RHO-1::Venus (day 4, right panel) normalized to actin levels. (B) Immunoblot and quantification of sample buffer insoluble fractions from lysates shown in (A, right panel). Samples in (A) were diluted 1:2 compared to samples in (B). See also S1 Data.
(EPS)

**S7 Fig. Safety mechanism acts independently of compensatory up-regulation of core PQC pathways.** (A) Reduced pharyngeal KIN-19 aggregation upon proteasome inhibition (*pas-6* RNAi) is not restored by macroautophagy inhibition (*atg-18(-)*). Changes in KIN-19::mEOS2 aggregation evaluated over time in the worm population. (B) Reduced pharyngeal RHO-1 aggregation upon chaperone impairment (*hsf-1(-)*) is not restored by macroautophagy inhibition (*atg-18* RNAi). Changes in RHO-1::tagRFP aggregation at day 1 in the worm population.

*P*-values determined by Fisher's exact test and Chi-square test. **$p < 0.01$, ***$p < 0.001$, ****$p < 0.0001$. See also S1 Data for number of animals evaluated and statistics.
(EPS)

**S8 Fig. Activation of the safety mechanism requires a proteostasis challenge.** (A) Flowchart for the analysis of RNA sequencing data showing steps to identify genes up-regulated in response to PQC disruption and protein aggregation in the pharynx but not in response to PQC disruption and protein aggregation in the body-wall muscles or with the fluorescent marker alone. See also S2 Data. (B) Knockdown by RNAi of genes identified by RNA-seq as selectively up-regulated by PQC failure in transgenic animals with pharyngeal RHO-1 and KIN-19 aggregation. Quantification of RHO-1::tagRFP aggregation at day 1 without PQC failure (B). Ordinal logistic regression analysis comparing all RNAi treatments to the control RNAi (B), see S3 Data for repeats. (C–F) Heat stress at 30˚C for 24 h induces *Ppals-5::GFP* expression in the pharynx and intestine (E, F) compared to standard growth conditions at 20˚C (C, D). (F) Magnification of the head region (outlined) shows localization of GFP induced by heat stress in the pharynx (*Pmyo-2::mCherry* in top image and outlined in bottom image). Single plane representative confocal images of day 1 adults are shown. DIC: differential interference contrast. Scale bar: 100 μm (C, E), 30 μm (D, F).
(EPS)

**S1 Table. Genes up-regulated in response to PQC disruption and protein aggregation in the pharynx.**
(DOCX)

**S1 Data. Source data.** Excel file with source data for figures.
(XLSX)

**S2 Data. Differentially expressed genes upon PQC failure.** Excel file, sheet 1 shows all differentially expressed genes, sheet 2 shows genes significantly regulated in transgenics with pharyngeal KIN-19 or RHO-1 aggregation and not in controls, sheet 3 shows safety mechanism related candidates up-regulated in both transgenics with pharyngeal KIN-19 and RHO-1 aggregation, and sheet 4 shows enrichment analysis of safety mechanism-related components regulated by microbes.
(XLSX)

**S3 Data. Investigation of safety mechanism related components by RNAi.** Excel file, sheet 1 shows results of 4 repeats in *hsf-1(-)* background and sheet 2 shows results of 2 repeats in wild-type background.
(XLSX)

**S1 Raw Images. Original images for blots.** Blots corresponding to S2C Fig; blots corresponding to S6A Fig (Left panel); related to S6A Fig (left panel), additional blot showing band specificity of KIN-19::mEOS2 detected by anti-CK1 antibody in CF3706 transgenics and not in N2 wild-type, and corresponding anti-actin loading control blot; blots corresponding to S6A Fig (right panel); blot corresponding to S6B Fig.
(PDF)

# Acknowledgments

We thank Xiaochen Wang for sharing *scav-3(qx193)* mutants and SCAV-3 overexpressing animals and Manuel Schölling for help with statistics based on the ordinal logistic regression model. We thank Nino Läubli and Sofia Kapsiani for help with the FLIM analysis and Emily

Troemel for advice and critical input. Some strains were provided by the CGC, which is funded by NIH Office of Research Infrastructure Programs (P40 OD010440).

## Author Contributions

**Conceptualization:** Della C. David.

**Data curation:** Christian Rödelsperger, Della C. David.

**Formal analysis:** Raimund Jung, Chyi Wei Chung, Harry C. Jones, Maximilian A. Thompson, Christian Rödelsperger, Gabriele S. Kaminski Schierle, Della C. David.

**Funding acquisition:** Gabriele S. Kaminski Schierle, Ralf J. Sommer, Della C. David.

**Investigation:** Raimund Jung, Marie C. Lechler, Ana Fernandez-Villegas, Chyi Wei Chung, Harry C. Jones, Yoon Hee Choi, Maximilian A. Thompson, Waltraud Röseler, Della C. David.

**Methodology:** Raimund Jung, Marie C. Lechler, Ana Fernandez-Villegas, Chyi Wei Chung, Harry C. Jones, Yoon Hee Choi, Maximilian A. Thompson, Christian Rödelsperger, Waltraud Röseler, Della C. David.

**Project administration:** Della C. David.

**Supervision:** Gabriele S. Kaminski Schierle, Ralf J. Sommer, Della C. David.

**Validation:** Raimund Jung, Ana Fernandez-Villegas, Harry C. Jones, Della C. David.

**Visualization:** Yoon Hee Choi, Della C. David.

**Writing – original draft:** Marie C. Lechler, Ralf J. Sommer, Della C. David.

**Writing – review & editing:** Della C. David.

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
