## [Editor Report · Decision Letter 0]

11 Jan 2023

Dear Dr David, 

Thank you for submitting via Review Commons the revised version of your manuscript entitled "Tissue-specific safety mechanism results in opposite protein aggregation patterns during aging" for consideration as a Research Article by PLOS Biology. Thank you also for your patience while we completed our editorial process over the Christmas holidays, and please accept my apologies for the delay in providing you with our decision.

Your manuscript has now been evaluated by the PLOS Biology editorial staff as well as by an academic editor with relevant expertise and I am writing to let you know that we would like to send your submission back to the original reviewers.

However, before we can do this, we need you to complete your submission by providing the metadata that is required for full assessment. To this end, please login to Editorial Manager where you will find the paper in the 'Submissions Needing Revisions' folder on your homepage. Please click 'Revise Submission' from the Action Links and complete all additional questions in the submission questionnaire.

Once your full submission is complete, your paper will undergo a series of checks in preparation for peer review. After your manuscript has passed the checks it will be sent out for review. To provide the metadata for your submission, please Login to Editorial Manager (https://www.editorialmanager.com/pbiology) within two working days, i.e. by Jan 13 2023 11:59PM.

Kind regards,

Ines

--

Ines Alvarez-Garcia, PhD

Senior Editor

PLOS Biology

---

## [Decision Letter · Decision Letter 1]

14 Mar 2023

Dear Dr David,

Thank you for your patience while we considered your revised manuscript from Review Commons entitled "Tissue-specific safety mechanism results in opposite protein aggregation patterns during aging" for consideration as a Research Article at PLOS Biology. Please also accept my apologies again for the delay in providing you with our decision. Your revised study has now been evaluated by the PLOS Biology editors, an Academic Editor and by the three original reviewers. 

The reviews are attached below. You will see that the reviewers appreciate the improvements you have made in the manuscript and both Reviewers 1 and 3 are now satisfied and recommend acceptance. Reviewer 2, however, raises several remaining points that would need to be addressed, including adding some staining and quantifications. This reviewer also thinks you should clearly state the conceptual advance of this manuscript in comparison with your recent publication in eLife. After discussions with the Academic Editor, s/he has also argued that day 7 worms are not considered old, thus we would also like you to insert ‘mid-life’ in several places in the text to refer to these worms and acknowledge that point. Please also note in the text that the pharynx muscles start to fail well before worms are considered "old", thus the tissues considered could age at different times.

In light of the reviews, we are pleased to offer you the opportunity to address the remaining points from the reviewers in a revision that we anticipate should not take you very long. In addition, we would like you to convert the manuscript into a Short Report format as we do feel it fits better in that format given that the potential components of the compensatory pathway described remain unknown. Please note that the main difference between the two formats is that Short Reports have a maximum of four main figures, so you will need to convert one of them into a Supplementary figure.

Once you resubmit, we will assess your revised manuscript and your response to the reviewers' comments with our Academic Editor aiming to avoid further rounds of peer-review, although might need to consult with the reviewers, depending on the nature of the revisions.

**IMPORTANT - SUBMITTING YOUR REVISION**

3. Resubmission Checklist

a) *PLOS Data Policy*

b) *Published Peer Review*

c) *Blurb*

Please also provide a blurb which (if accepted) will be included in our weekly and monthly Electronic Table of Contents, sent out to readers of PLOS Biology, and may be used to promote your article in social media. The blurb should be about 30-40 words long and is subject to editorial changes. It should, without exaggeration, entice people to read your manuscript. It should not be redundant with the title and should not contain acronyms or abbreviations. For examples, view our author guidelines: https://journals.plos.org/plosbiology/s/revising-your-manuscript#loc-blurb

Sincerely,

Ines

--

Ines Alvarez-Garcia, PhD

Senior Editor

PLOS Biology

Reviewers' comments

Rev. 1:

The authors have addressed my concerns.

Rev. 2:

Previous comment from reviewer:

**Major comments:**

1. The three major pathways of protein quality control: proteasome, macroautophagy and chaperones were studied by depleting 6 genes in total using systemic RNAi. Meaning these genes are depleted in the entire animal. There is zero discussion on trans-tissue signalling that could lead to a signalling between different tissues and an adjustment to the RNAi-mediated depletion of these pathways. How do the authors know that a RNAi-mediated depletion of e.g. pbs-3 indeed only affects the proteasome (in the studied tissues and everywhere else)? Why did the authors not express sensors for the respective pathways in the pharynx (they already exist for body wall muscle) and first establish their RNAi-mediated approach?

Answer by authors:

We agree with the reviewer that cell non-autonomous coordination, including transcellular signalling for example involving the up-regulation of chaperones, plays an important role in proteostasis. Impairment of PQC in one tissue could lead to a compensatory upregulation of PQC in another tissue. However, our results in Fig. 4A and 4B show that inhibition of several PQC pathways together does not eliminate the safeguard response.

Response from reviewer:

This does not address the raised point that the selected genes (for knockdown) affect one and only one specific pathway. To ensure that the model and tools used are valid, one would need to first express reporters for the gene or function. Models for the proteasome, autophagy and protein folding in C.elegans have been established by the Holmberg, Hansen and Hartl labs, respectively. Using those models, one could then apply a knockdown by either systemically depleting the gene or better yet by using a tissue-specific knockdown approach. Next, the knockdown efficiency needs to be established (see point 2) and the effect on the selected protein quality control (PQC) pathway as well as the other remaining pathways (to ensure specificity for the PQC pathway) have to be tested.

This is important as some proteins e.g. Hsc70/HSP-1 are involved in protein folding, but also in chaperone-mediated autophagy and macroautophagy as well as targeting of protein substrates to the proteasome. So it is not as easy to dissect the different branches of the PQC and hence requiring a validation of the employed tools and models. And this does not even yet take into account the possibility of a crosstalk between the PQC pathways to adjust each other in response to an inhibition of one branch.

Answer by authors:

To avoid possible adjustment to RNAi-mediated depletion of the pathways, we used available mutants to impair HSF-1 and macroautophagy throughout the animal. As impairment of the proteasome during development is lethal, we could not use null mutants for this pathway. Yet we are confident that the proteasome is effectively inhibited in the pharynx as we found a significant increase in PAB-1 aggregation in the pharynx upon proteasome inhibition by RNAi (Fig. S3).

Response from reviewer:

This observation is not an evidence of the effectiveness of the RNAi of proteasomal genes. If at all, the protein levels of PAB-1 need to be tested first.

Answer by authors:

Together, these results demonstrate that reduced aggregation of RHO-1 and KIN-19 in the pharynx is not due to a compensatory upregulation of the core PQC pathways.

Response from reviewer:

See the comments above.

Answer by authors:

Instead our results show that aggregation of globular proteins during PQC impairment is avoided by macroautophagy-independent targeting to the lysosome and components of the intracellular pathogen response.

Response from reviewer:

How is this supposed to work? There are excellent imaging tools available to study the lysosomal pathway. These should be used to track the degradation of RHO-1 and KIN-19 in ideally time lapse imaging analyses to support the claim that these proteins are degraded by the lysosome yet in a macroautophagy-independent manner.

Answer by authors:

Still it remains possible that inter-tissue signalling is responsible for induction of this safety mechanism. Yet, a cell autonomous response is a more likely explanation because of the involvement of the intracellular pathogen response which would normally be activated in the digestive tract directly exposed to pathogen (see results with pals-5 in Fig.5 and S6). We have updated the discussion to include the possibility of cell non- autonomous regulation.

previous comment from reviewer:

2. Neither RNAi-specificity nor effectivity were assessed in the pharynx vs. body wall muscle. What are the remaining protein or at least mRNA levels of the 6 modulators upon RNAi? Immunolabelling could enable a tissue-specific assessment of the remaining levels.

Answer by authors:

We thank the reviewer for this insightful point. We replicated our RNAi results with loss of function mutations: atg-18(gk378) mutants have a full atg-18 knockout with a deletion that removes the start codon from atg-18. unc-51(e369) is a substitution that disrupts autophagic structures (doi: 10.1080/15548627.2014.1003478). The hsf-1(sy441) mutation introduces an early stop codon resulting in a truncated HSF-1 lacking its C-terminal transactivation domain. We have added a sentence in the result section to underline our use of mutants to confirm the RNAi results.

Response from reviewer:

The authors have used three mutants that affect 2 pathways (autophagy and induction of chaperones). The possibility of using inhibitors for e.g. the proteasome (MG132, epoxomicin, bortezomib) was not considered.

previous comment from reviewer:

3. It has not been addressed that the RNAi might not affect pharynx and body wall muscle alike. Why did they not employ tissue-specific depletion?

Answer by authors:

In addition to using mutants (see response to point 2), we made sure that RNAi is effective in the pharynx by controlling for GFP fluorescence levels after subjecting the animals to gfp RNAi. Using a similar experimental setup as performed for the RNAi targeting the PQC, we showed that gfp RNAi efficiently reduces GFP levels to less than 25% of control levels at day 1 of adulthood (over 4 fold reduction) (new Fig. S1F).

Response from reviewer:

The analysis of RNAi of gfp does not address how efficient the pbs-3, hsf-1, atg-18 are depleted in the two muscle tissues.

previous comment from reviewer:

4. Does an over-expression of the respective gene (pbs-3, hsf-1 etc) rescue the observed effect?

Answer by authors:

We tested how enhancing PQC influences protein aggregation in the pharyngeal muscles. We found that HSF-1 overexpression or enhancing proteasome activity by RPN-6 overexpression alleviated aggregation pharyngeal protein aggregation (new Fig. S2). Thus, these core PQC systems are normally responsible for preventing protein aggregation localized in the pharynx. These results support our conclusions that a safety mechanism is triggered because core PQC impairment renders the pharynx vulnerable to protein aggregation.

Response from reviewer:

Why are - like in almost all data sets that were added during revision - only secondary data (plotted graphs) shown?

RPN-6 and HSF-1 overexpression has been tested. An autophagy analogue is missing. One could have used a chemical activation of the autophagy pathway.

There is no demonstration that RPN-6 and HSF-1 are in fact overexpressed in the pharynx. This needs to be demonstrated.

previous comment from reviewer:

5. Not all chaperones are regulated by HSF-1. So the claim the depletion of hsf-1 could be used as tool to suppress the activity of molecular chaperones is not valid. Why not instead analyse the aggregation-propensity upon modulation of selected major chaperones such as hsp-1, daf-21 or hsp-16?

Answer by authors:

We appreciate this point, although we note that HSF-1 controls the expression of the major chaperones in basal conditions. In particular, Brunquell et al. (doi: 10.1186/s12864-016- 2837-5) demonstrate that hsf-1 inhibition by RNAi leads to the downregulation of hsp-1, hsp- 12, hsp-16.1, hsp-16.11, hsp-16.2, hsp-16.41, hsp-16.48, hsp-16.49 and hsp-3. Moreover, our RNA seq data (Data file S2) showed that inhibition of hsf-1 by RNAi (compared to empty vector alone) leads to the downregulation of hsp-70, hsp-110, hsp16.48, hsp-16.49, hsp-16.41 and hsp-16.2. O'Brien et al. show that hsf-1RNAi treatment reduced levels of daf-21 (10.1016/j.celrep.2018.05.093). Together these data establish that knockdown of HSF-1 is an efficient way to deplete a large selection of chaperones. Nonetheless, following the recommendation of the reviewer, we have now evaluated the role of hsp-1 by RNAi and show that inhibiting this major chaperone leads to reduced aggregation in the pharynx (new Fig S1E). These exciting new results are consistent with our findings in hsf-1-defective animals, showing that depletion of a single chaperone is also sufficient to trigger the safety mechanism.

Response from reviewer:

What is the reason that the authors only show secondary data for hsp-1 knockdown (Fig S1e). And why was only RHO-1 assessed?

And as above, what are the knockdown efficiencies of hsp-1 in the pharynx?

Were other chaperones tested as well?

previous comment from reviewer:

6. The observed reduced fluorescence levels in Fig. 1C+E upon depletion of atg-18 show an overall reduction in protein levels and not just of foci. How do the authors exclude the possibility that the overall levels of the 3 studied proteins change and not just the fraction of aggregated proteins? This is important as it is established that protein levels correlate with their aggregation-propensity.

Answer by authors:

We agree with the important point that aggregation can be affected by total protein levels. However, we can exclude an overall reduction in protein levels as inhibition of PQC (pbs-3, unc-51 and hsf-1) leads to accelerated PAB-1 aggregation in the pharynx (Fig. S3).

Response from reviewer:

This does not address the concern. What is needed is a quantification of the protein levels. The fluorescence levels already indicate a reduction in the protein abundance. This should be confirmed by e.g. immunostaining.

Answer by authors:

Moreover, we can also exclude translation inhibition in response to atg-18 or pbs-3 depletion as total levels of the transcriptional reporter mEOS alone under the control of the kin-19 promoter are not reduced by these treatments (Fig. S4B).

Response from reviewer:

This response does not address the concern either.

previous comment from reviewer:

7. I miss an analysis of the nature of the foci of the three proteins in particular upon modulation of the protein quality control pathways. The authoring lab has used in previous studies the FLIM technique. Why not here? This would add some quantitative data to this study that otherwise relies heavily on an assessment of nematodes with detectible foci. FRAP as well as biochemical fractionation would be another option.

Answer by authors:

We thank the reviewer for this helpful suggestion. We have now performed FLIM to analyse the nature of protein aggregation during PQC impairment. We show an increase in fluorescence lifetime in the pharynx of animals with PQC disruption (new Fig. 2A). This reveals that the safety mechanism triggered effectively lowers the accumulation of amyloid- like aggregates.

Response from reviewer:

I appreciate that the authors used the FLIM technique. The reason that I asked for either FRAP or FLIM was to validate the foci as aggregated species. A much higher resolution and magnification is needed to differentiate between foci and diffuse proteins to then assess them by FLIM. Alternatively, use FRAP to analyze the mobility of the foci.

previous comment from reviewer:

8. The photo-conversion approach should actually be extremely useful to identify the fate of the photo-converted protein. If it is indeed getting degraded, a treatment with UPS or autophagy inhibitors should stabilise the protein. A chaperone-mediated disaggregation should lead to an accumulation of resolubilised soluble protein.

Answer by authors:

We thank the reviewer for highlighting the photoconversion approach to study the dynamics of protein aggregation. We followed the fate of the photoconverted red-labelled aggregates after PQC failure to investigate disaggregation. Upon inhibition of the proteasome or macroautophagy, the low rate of removal of KIN-19 aggregates was similar to control RNAi and moderately enhanced for RHO-1 aggregates (Fig. 3A to C). Therefore, disaggregation does not play a major role in the safety mechanism. In contrast, excitingly we observed a large reduction in the rate of new green-labelled aggregate formation in all conditions. As the reviewer recommends, we investigated whether soluble proteins are being degraded by measuring the total fluorescence after triggering the safety response. Indeed after 48h, we observed less total levels of newly synthesized KIN-19::mEOS while inhibiting the proteasome (pbs-3 RNAi) or inhibiting macroautophagy (atg-18 RNAi). Together, this evidence demonstrates that the KIN-19::mEOS is degraded before it assembles into aggregates.

Response from reviewer:

Such an analysis (ideally depicted in a time lapse manner) would allow to track individual foci to literally see if they get degraded or resolubilized in the different genetic backgrounds.

The depicted plotted graphs (no imaging raw data) do not allow an assessment.

previous comment from reviewer:

9. I strongly recommend to analyse the protein aggregate clearance a global / proteomic and unbiased approach. An analysis of these three previously studied proteins might not be representative to draw conclusion on the protein quality control strategies for an entire tissue.

Answer by authors:

We agree that it would be desirable to perform proteomic-wide analyses. Unfortunately, it is not possible to dissect out the pharynx in C. elegans. Moreover, the techniques needed to perform tissue-specific proteomics, e.g. proximity labelling (doi: 10.1126/sciadv.1602426) or cell-selective bioorthogonal noncanonical amino acid tagging (doi: 10.1073/pnas.1421567112), have only been recently adapted to C. elegans. Therefore, we feel that the method development needed to adjust these methods to characterize tissue- specific protein aggregation is out of the scope of the current work and we plan to do this in the future.

Response from reviewer:

If an unbiased approach is not possible, the authors should avoid generalizing claims and only refer to RHO-1, KIN-19 and PAB-1.

previous comment from reviewer:

10. On page 4 last paragraph: the authors state that they used RNAseq of "animals with pharyngeal protein aggregates but not induced PQC impairment in animals with body wall muscle aggregates" without stating the actual strains or RNAi conditions. This is not acceptable.

Answer by authors:

Extensive information related to the strains used and RNAi conditions can be found in the Materials and Methods section "Animal collection for RNA sequencing" and Figure S5 describes the analysis procedure of the RNA sequencing data. In addition, we have now included in the result section more detail related to the experimental design.

Response from reviewer:

Ok.

Rev. 3: Richard Sifers – note that this reviewer has signed his review

The authors successfully responded to my original comments. Their conclusions are now intact because they are supported by additional experimentation and more accurate explanation of their findings. In my opinion, the revised manuscript describes an interesting and important study worthy of publication in PLOS Biology.

---

## [Editor Report · Decision Letter 2]

14 Jul 2023

Dear Dr David,

Thank you for your patience while we considered your revised manuscript entitled "Safety mechanism enables tissue-specific resistance to protein aggregation during aging." for publication as a Short Report at PLOS Biology. This revised version of your manuscript has been evaluated by the PLOS Biology editors and the Academic Editor.

Based on our Academic Editor's assessment of your revision, we are likely to accept this manuscript for publication, provided you satisfactorily address a couple of points:

- The Academic Editor would like you to add a couple of sentences to the manuscript comparing your data to the HSF-1 data shown in Brunquell et al study for the benefit of the readers.

- We would also like you to consider a suggestion to improve the title:

"A safety mechanism enables tissue-specific resistance to protein aggregation during aging in C. elegans"

We expect to receive your revised manuscript within two weeks. 

*Published Peer Review History*

*Press*

Sincerely,

Ines

--

Ines Alvarez-Garcia, PhD

Senior Editor

PLOS Biology

---

## [Editor Report · Decision Letter 3]

1 Aug 2023

Dear Dr David,

My name is Luke Smith - I am an editor at PLOS Biology and I am writing on behalf of my colleague Ines who is away on vacation this week. I am handling your manuscript while Ines is away. Thank you for the submission of your revised Short Report "A safety mechanism enables tissue-specific resistance to protein aggregation during aging in C. elegans" for publication in PLOS Biology, and thank you for addressing our last editorial requests in this revision. On behalf of my colleagues and the Academic Editor, Heidi A. Tissenbaum, I am pleased to say that we are satisfied by the changes made and that we can in principle accept your manuscript for publication. Please note, however that before publication we will need you to address any remaining formatting and reporting issues. These will be detailed in an email you should receive within 2-3 business days from our colleagues in the journal operations team; no action is required from you until then. Please note that we will not be able to formally accept your manuscript and schedule it for publication until you have completed any requested changes.

PRESS

Sincerely, 

Luke

Lucas Smith, PhD

Senior Editor

PLOS Biology

lsmith@plos.org

on behalf of

Ines Alvarez-Garcia, PhD

Senior Editor

PLOS Biology
